# Lansoprazole Ameliorates Isoniazid-Induced Liver Injury

**DOI:** 10.3390/ph17010082

**Published:** 2024-01-08

**Authors:** Eri Wakai, Takashi Shiromizu, Shota Otaki, Junko Koiwa, Satoshi Tamaru, Yuhei Nishimura

**Affiliations:** 1Department of Integrative Pharmacology, Mie University Graduate School of Medicine, Tsu 514-8507, Mie, Japan; wakaie@pharma2.med.osaka-u.ac.jp (E.W.); tshiromizu@med.mie-u.ac.jp (T.S.); 320018@m.mie-u.ac.jp (S.O.); j-koiwa@med.mie-u.ac.jp (J.K.); 2Mie University Research Center for Cilia and Diseases, Tsu 514-8507, Mie, Japan; 3Clinical Research Support Center, Mie University Hospital, Tsu 514-8507, Mie, Japan; tamaru3@med.mie-u.ac.jp

**Keywords:** drug-induced liver injury, drug repositioning, gene expression signature, connectivity map, FDA adverse event reporting system, zebrafish, electronic health records

## Abstract

Isoniazid is a first-line drug in antitubercular therapy. Isoniazid is one of the most commonly used drugs that can cause liver injury or acute liver failure, leading to death or emergency liver transplantation. Therapeutic approaches for the prevention of isoniazid-induced liver injury are yet to be established. In this study, we identified the gene expression signature for isoniazid-induced liver injury using a public transcriptome dataset, focusing on the differences in susceptibility to isoniazid in various mouse strains. We predicted that lansoprazole is a potentially protective drug against isoniazid-induced liver injury using connectivity mapping and an adverse event reporting system. We confirmed the protective effects of lansoprazole against isoniazid-induced liver injury using zebrafish and patients’ electronic health records. These results suggest that lansoprazole can ameliorate isoniazid-induced liver injury. The integrative approach used in this study may be applied to identify novel functions of clinical drugs, leading to drug repositioning.

## 1. Introduction

Drug-induced liver injury (DILI) is the most common cause of acute liver failure in developed countries [1]. The diagnosis, treatment, and prevention of DILI remain a challenge [2,3,4].

Isoniazid (INH), a first-line drug in antitubercular therapy [5], is one of the most commonly implicated drugs in DILI [6]. Elevation of liver enzymes may occur in up to 20% of patients receiving INH monotherapy or a combination antitubercular therapy [7]. The elevation of liver enzymes usually resolves with INH discontinuation. However, INH can cause acute liver failure, leading to death or emergency liver transplantation [8]. The mechanisms underlying INH-induced liver injury (IILI) include oxidative stress, mitochondrial dysfunction, endoplasmic reticulum (ER) stress, bile acid transport imbalance, and immune response [9]. Therapeutic approaches for the prevention of IILI are yet to be established. Therefore, it is essential to fully elucidate the mechanisms of IILI and identify drugs that can prevent it.

The mouse diverse panel (MDP), a panel of inbred mouse strains with genetic diversity, had been successfully used to elucidate susceptibility to chemicals, including INH [10,11]. MDP could also be used to identify genes that were differentially expressed between phenotypes of interest [10,12]. A gene list that is differentially expressed between phenotypes, also called a gene expression signature (GES), can be utilized to predict chemicals that may cause the same or opposite pattern of a GES by implementing the concept of a connectivity map [13,14,15,16].

Zebrafish have been used to analyze a variety of liver diseases [17,18,19]. Although the zebrafish liver has a tissue architecture that is different from that of humans, the cell types and the functions of the zebrafish liver are similar to those of humans [17,20]. Zebrafish have been successfully utilized to study liver injury caused by hepatotoxicants, including INH [18,21,22,23].

In this study, we identified a GES associated with IILI using public MDP transcriptome data, predicted lansoprazole (LPZ), a proton pump inhibitor that is widely used for the management of acid-related diseases, as a protectant against IILI, and we validated its effects using zebrafish and electronic health records, providing the utility of zebrafish in an integrative approach for drug discovery.

## 2. Results

### 2.1. Identification of the Gene Expression Signature Associated with IILI

This study used a public transcriptome dataset using MDP while focusing on the susceptibility to IILI [10] to identify the GES associated with IILI. Three IILI-susceptible inbred mouse strains (BUB/BnJ, DBA/2J, and LG/J) were selected in the MDP. The transcriptome analysis of the IILI-susceptible strains revealed that 135, 15, and 89 genes were significantly upregulated by INH in the livers of BUB/BnJ, DBA/2J, and LG/J mice, respectively, and 30 genes were upregulated in at least two of the three strains (Figure 1A and Appendix A). The transcriptome analysis also revealed that 64, 183, and 129 genes were significantly downregulated by INH in the livers of BUB/BnJ, DBA/2J, and LG/J mice, respectively, and that 78 genes were downregulated in at least two of the three strains (Figure 1B and Appendix A).

In the MDP, three inbred mouse strains, C57BR/cdJ, Non/ShiLtj, and PWK/PhJ, were selected as resistant to IILI. The transcriptome analysis of the IILI-resistant strains revealed that 124, 53, and 94 genes were significantly upregulated by INH in the livers of C57BR/cdJ, Non/ShiLtj, and PWK/PhJ mice, respectively, and 20 genes were upregulated in at least two of the three strains (Figure 1C and Appendix A). The transcriptome analysis also revealed that 41, 79, and 205 genes were significantly downregulated by INH in the livers of C57BR/cdJ, Non/ShiLtj, and PWK/PhJ mice, respectively, and 26 genes were downregulated in at least two of the three strains (Figure 1D and Appendix A).

The comparison of the upregulated and downregulated genes in the sensitive and resistant strains revealed 24 upregulated and 66 downregulated genes in the sensitive strains but not in the resistant strains (Figure 1E,F, Appendix A). The 24 upregulated and 66 downregulated genes were defined as a GES associated with IILI. Gene ontology analysis revealed various functions enriched in the GES, including “Unfolded Protein Response” in the upregulated GES and “negative regulation of tumor necrosis factor production” and “PPAR signaling pathway” in the downregulated GES (Table 1). These functions have been related to the hepatotoxicity of INH [9,21,22,24].

### 2.2. Lansoprazole Suppressed INH-Induced Apoptosis in Zebrafish Liver

Cmap and Lincs Unified Environment (Clue) was used to identify therapeutic drugs against IILI. Clue is a platform for predicting chemicals that could cause gene expression changes similar to or opposite to those of a GES [25]. We hypothesized that therapeutic drugs against IILI may cause gene expression changes opposite to those associated with IILI. Using the GES as the input, Clue identified 644 chemicals that may cause gene expression changes opposite to the GES associated with IILI. The FDA Adverse Event Reporting System (FAERS) database [26] was used to identify clinical drugs that could potentially suppress IILI. The FAERS had 10,285 patients who experienced IILI. The analysis of the patients identified 57 co-administered drugs that significantly reduced the reporting odds ratio of IILI. Thirteen clinical drugs were identified using Clue and FAERS (Figure 2A), of which we were interested in LPZ because previous studies demonstrated its protective action in various rat models of liver injury [27,28,29,30].

To examine the effects of LPZ on IILI, Tg (pck1:tagGFP-DEVD-tagRFP), a transgenic zebrafish line expressing tagGFP-DEVD-tagRFP in the liver [23], was used. When caspase 3 was not activated in the liver, the DEVD linker between tagGFP and tagRFP was not digested. In this case, excitation at the absorbance wavelength of tagGFP caused emission by tagRFP owing to the Förster resonance energy transfer (FRET). When caspase 3 was activated in the liver, the DEVD linker was digested, decreasing the FRET. In vivo fluorescence images of zebrafish using a narrow filter (Ex/Em: 460–480/505–530 nm) and a wide filter (Ex/Em: 460–480/567–647 nm) were used to assess apoptosis in the liver (Figure 2B). The fluorescence intensity of the livers of zebrafish treated with INH increased in the narrow filter, suggesting that INH treatment activated caspase 3 in the liver. Cotreatment with LPZ and INH suppressed the increase in fluorescence of the narrow filter. Quantitative analysis using the fluorescence intensity ratio in the narrow and wide filters revealed that cotreatment with LPZ significantly suppressed INH-induced activation of caspase 3 in the zebrafish liver (Figure 2C).

To analyze the mechanism underlying the effects of LPZ against INH-induced apoptosis in zebrafish livers, quantitative PCR (qPCR) analysis was performed. The expression of two genes, DNA damage-inducible transcript 3 (DDIT3, also called CHOP) and tumor necrosis factor-α (TNFA), was the focus. DDIT3 is a representative marker for ER stress [31]. Increases in ER stress and TNFA expression are involved in the mechanisms underlying IILI [9,21,22]. The qPCR analysis revealed that LPZ significantly suppressed the expression of *ddit3* and *tnfa* induced by INH (Figure 2D), suggesting that LPZ may protect against IILI at least partly by suppressing DDIT3 and TNFA.

### 2.3. Lansoprazole Was Protective against IILI in Patients

To examine the effect of LPZ on IILI, a retrospective analysis was performed using the clinical data of patients who received INH in the electronic health records of Mie University Hospital. Based on the inclusion and exclusion criteria, 256 patients were selected for INH treatment. Patient characteristics are shown in Table 2. Of the 256 patients, 30 received LPZ. The age and duration of INH administration were significantly higher in patients treated with LPZ. In contrast, AST and ALT levels after INH treatment were significantly lower in patients treated with LPZ. These results suggest that cotreatment with LPZ suppresses IILI in patients.

## 3. Discussion

In this study, we predicted that LPZ was protective against IILI through an integrative in silico analysis using public transcriptome data from MDP, with a focus on IILI, and a public self-reporting system for adverse drug events. We confirmed the hepatoprotective effects of LPZ using zebrafish and patients’ electronic health records.

The hepatoprotective effects of LPZ have been demonstrated in several rat models. In an acute liver injury model induced by d-galactosamine and lipopolysaccharide, LPZ inhibited the activity of nuclear factor-kappa B and suppressed the increases in TNFA, inducible nitric oxide synthase, and cytokine-induced neutrophil chemoattractant-1 expression in the liver [27]. Consistently with this report, we demonstrated that LPZ suppressed the increase in *tnfa* expression induced by INH. Increased expression of TNFA in IILI has been shown [9,22]. These findings suggest that LPZ may partially ameliorate IILI by suppressing TNFA expression.

We used gene ontology analysis to demonstrate that the unfolded protein response (UPR) was enriched in the upregulated GES for IILI. UPR is tightly linked to ER stress associated with IILI [32,33]. Previous zebrafish studies demonstrated that 4-phenylbutyrate, an ER stress inhibitor, ameliorates liver injury induced by INH and lipopolysaccharides [22]. Consistently with these reports, we demonstrated that LPZ suppressed the increased expression of *ddit3* induced by INH. These studies suggest that the suppression of ER stress may also be involved in the protective mechanism of LPZ against IILI. The mechanisms of how LPZ can suppress ER stress remain unknown.

The activation of nuclear factor erythroid-derived 2-like 2 (NRF2) has also been demonstrated as a protective mechanism of LPZ against non-alcoholic steatohepatitis in model rats induced by a choline-deficient amino-acid-defined diet [28] and acute liver damage in a rat model induced by thioacetamide [29]. LPZ activates NRF2 signaling pathways by phosphorylating p38 mitogen-activated protein kinase [30]. Activation of NRF2 signaling increases the expression of antioxidant genes in the liver [34]. Oxidative stress has been demonstrated as a critical mechanism underlying IILI [9,21,35]. NRF2 activation may be involved in the protective effects of LPZ against IILI.

We showed that co-treatment of LPZ suppressed the activation of caspase 3 induced by INH in zebrafish liver. It was demonstrated that caspase 3 was activated by INH in a human hepatocellular carcinoma line [36] and that suppression of apoptosis was involved in the hepatoprotective effects of chemicals against INH-induced injury in a human liver cell line and rat liver [37,38]. LPZ suppressed apoptosis in an acute liver injury rat model [27]. These reports suggest that LPZ may suppress apoptosis induced by INH in the human liver.

We also demonstrated that the increase in ALT and AST levels and the duration of INH treatment in patients treated with INH and LPZ were significantly lower and longer, respectively, than in patients treated with INH without LPZ. These findings suggest that the co-administration of LPZ with INH may decrease the risk of IILI and increase the possibility of completion of anti-tuberculosis therapy using INH. It should be noted, however, that the number of patients treated with INH and LPZ (*n* = 30) is about seven times smaller than that of patients treated with INH but not LPZ (*n* = 226). A more significant number of patients treated with INH and LPZ should be included to reduce the power imbalance and the sensitivity to outliers. It should also be mentioned that INH can inhibit cytochrome P450 family 2 subfamily C member 19 (CYP2C19) [39,40], a major phase I enzyme that metabolizes LPZ [41]. Although LPZ can be metabolized by cytochrome P450 family 3 subfamily A member 4 [41], the pharmacokinetics of LPZ co-treated with INH remain unknown. Further studies are required to fully elucidate the mechanism underlying the protective effect of LPZ against IILI and its impact on the prognosis of patients with tuberculosis.

In summary, we have demonstrated that LPZ ameliorates IILI by applying an integrative approach using in silico data, zebrafish, and real-world data, such as electronic health records and self-reporting data for adverse drug events. The integrative approach may be applicable to the identification of novel functions of clinical drugs, leading to drug repositioning. This approach relies on the availability of in silico and real-world data and assays using zebrafish that are relevant to the disease of interest. Data sharing and the development of various zebrafish-based assays have grown steadily [42,43], increasing the integrative approach’s applicability.

## 4. Materials and Methods

### 4.1. Ethics Statement

This study was conducted in accordance with the Declaration of Helsinki and approved by the Ethics Committee of Mie University Graduate School of Medicine and Faculty of Medicine (no. 3154 and no. H2021-105). This study was approved by the Institutional Animal Care and Use Committee of Mie University (no. 2020-19). All of the animal experiments conformed to the ethical guidelines established by the committee.

### 4.2. Transcriptome Analysis

The transcriptome dataset (GSE55489) [10] was obtained from the Gene Expression Omnibus [44]. We used the R/Bioconductor “affy” [45] and “RankProd” [46] packages to normalize the dataset and identify genes that were differentially expressed between the control and INH-treated groups using a false discovery rate threshold of 30%. Mouse gene symbols were converted into human orthologs using the Mouse Genome Database [47]. We used the Cytoscape ClueGO package [48] to identify the functions enriched in the GES for IILI.

### 4.3. Bio-/Chemoinformatic Analysis

Clue (https://clue.io/ (accessed on 15 November 2023)) was used to identify chemicals that could potentially generate the opposite GES pattern for IILI. Chemicals with median tau below –10 were identified as candidate chemicals that might suppress IILI.

The FAERS (https://www.fda.gov/drugs/questions-and-answers-fdas-adverse-event-reporting-system-faers/fda-adverse-event-reporting-system-faers-public-dashboard (accessed on 15 November 2023)) was used to identify clinical drugs that potentially suppressed IILI. The files recorded in the FAERS between 2015 and 2022 were downloaded. A GUI-based calculator was constructed using Python to analyze the effect of the co-administered drug (d) on the reporting odds ratio (ROR) for an adverse reaction (r) in patients treated with a drug causing an adverse event (drug A). The ROR was calculated from two-by-two contingency tables comprising the number of drug-A-treated cases that received d and had r (d1r1), received d and did not have r (d1r0), did not receive d and had r (d0r1), and did not receive d and did not have r (d0r0). The ROR was calculated using 95% confidence intervals. Drugs with a maximum 95% confidence interval (CI) <1 were identified as candidate drugs that may suppress IILI.

### 4.4. Compounds

INH and LPZ were purchased from the Tokyo Chemical Industry (Tokyo, Japan). Stock solutions of LPZ were prepared in dimethyl sulfoxide (Fujifilm Wako Pure Chemical, Komono-cho, Japan) at a concentration that was 1000-fold higher than the final concentration used in the experiments. The stock solutions were then diluted 1000-fold in 0.3× Danieau’s solution (19.3 mM NaCl, 0.23 mM KCl, 0.13 mM MgSO_4_, 0.2 mM Ca[NO_3_]_2_, and 1.7 mM HEPES [pH 7.2]). INH powder was dissolved in 0.3× Danieau’s solution to the final desired concentration.

### 4.5. Zebrafish Husbandry

Zebrafish were maintained according to standard methods, as described previously [49,50]. Briefly, zebrafish were raised at 28.5 ± 0.5 °C with a 14 h/10 h light/dark cycle. We obtained embryos through natural mating, cultured the embryos in 0.3× Danieau’s solution until 7 dpf, and then raised them in a recirculating system for housing zebrafish.

### 4.6. In Vivo Fluorescence Imaging Using Tg (tagGFP-DEVD-tagRFP)

Mature Tg (pck1:tagGFP-DEVD-tagRFP) were mated, and the resulting embryos were transferred to 6-well plates (20 embryos per well) and treated with the test chemicals at the indicated concentrations from 4 to 5 dpf (24 h). After treatment, the larvae were transferred to fresh 0.3× Danieau’s solution containing 2-phenoxyethanol (500 ppm) for anesthesia and then transferred to a glass-bottom dish (Cell View Cell Culture Dish, four compartments, Greiner Bio-One, Austria) along with the medium. The zebrafish were then observed using an inverted fluorescence microscope (Axio Observer, Zeiss, Oberkochen, Germany) with the following filters: wide filter (Ex/Em 460–480/567–647 nm) and narrow filter (Ex/Em 460–480/505–530 nm). The fluorescence signals in the obtained images were quantified using the image-processing package Fiji [51]. Briefly, 16-bit images were imported into Fiji, and areas of wide and narrow fluorescence above the threshold (mean gray value >4000) within the liver were measured. The ratios of the narrow and wide areas above the thresholds were then calculated.

### 4.7. qPCR Analysis

Total RNA was extracted from albino zebrafish treated with INH and LPZ using an RNeasy Mini Kit (Qiagen, Tokyo, Japan) according to the manufacturer’s instructions. cDNA was generated using the ReverTra Ace qPCR RT Kit (Toyobo, Osaka, Japan). qPCR was performed using Applied Biosystems StepOne (Thermo Fisher Scientific, Waltham, MA, USA) with THUNDERBIRD SYBR qPCR Mix (Toyobo). The thermal cycling conditions were the following: 95 °C for 1 min, followed by 40 cycles of 95 °C for 15 s, 60 °C for 15 s, and 72 °C for 45 s. The mRNA expression of *ddit3* and *tnfa* was normalized to that of actin beta 1 (*actb1*) to correct for variability in the initial template concentration and reverse transcription efficiency. The primer sequences used are listed in Appendix A.

### 4.8. Analysis of the Clinical Data of Patients Using Electrical Medical Records

The clinical data of patients (*n* = 256) who received INH at Mie University Hospital between January 2010 and 2018 were extracted from their electronic medical records. We excluded patients if their aspartate aminotransferase (AST) and alanine aminotransferase (ALT) were ≥100 IU/L, estimated glomerular filtration rate (eGFR) was <59 mL/min/1.73 m^2^, and serum creatinine (Scr) was ≥1.2 mg/dL before INH treatment. Clinical data before and after INH administration in patients treated with LPZ (*n* = 30) and without LPZ (*n* = 226) were compared.

### 4.9. Statistical Analysis

The Shapiro–Wilk normality test was used to examine the data distribution for each group. Fisher’s exact test or the Mann–Whitney U test was used for clinical data to explore the difference between the groups. For experimental data, ordinary one-way ANOVA with Tukey’s multiple-comparison test was used to examine the differences between groups. The Prism 10 software (GraphPad, San Diego, CA, USA) was used for all analyses.

## Figures and Tables

**Figure 1 pharmaceuticals-17-00082-f001:**
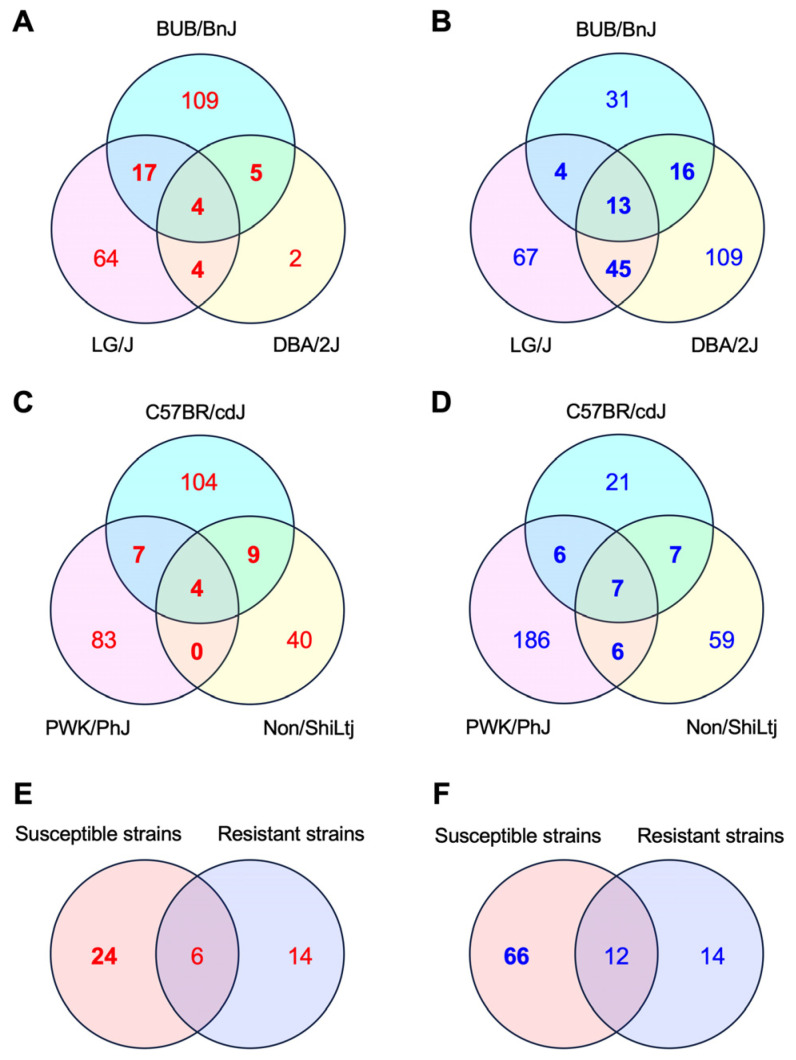
Comparative transcriptome analysis of the mouse livers that were susceptible or resistant to IILI. (**A**–**D**) Venn diagrams of unique and shared differentially expressed genes in untreated vs. INH-treated mouse livers that were susceptible (**A**,**B**) or resistant (**C**,**D**) to IILI. Differentially expressed genes were identified using a false discovery rate threshold of 30%. (**A**,**C**) show genes that were upregulated, whereas (**B**,**D**) show genes that were downregulated by the INH treatment. (**E**,**F**) Venn diagrams of genes that were upregulated (**E**) or downregulated (**F**) in at least two strains among those susceptible or resistant to IILI.

**Figure 2 pharmaceuticals-17-00082-f002:**
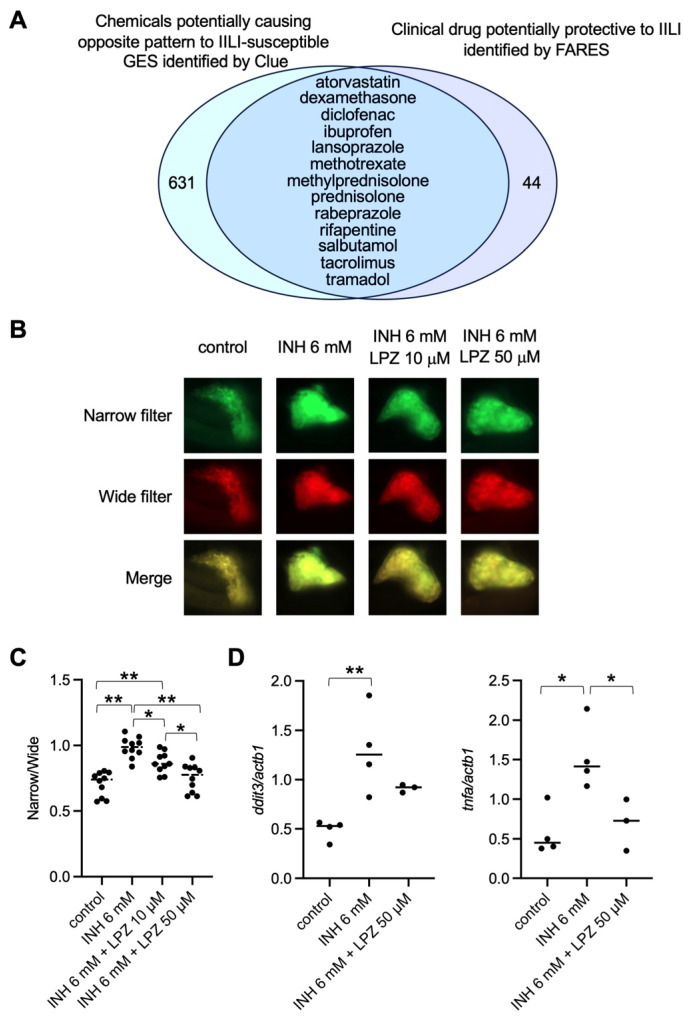
Identification of lansoprazole as a protectant against IILI using in silico and in vivo analyses. (**A**) Clue identified chemicals that potentially caused patterns opposite to those of the IILI-GES. The search for inverse signals via the FAERS identified clinical drugs that potentially worked against IILI. Thirteen clinical drugs were common in the three groups, including lansoprazole (LPZ). (**B**) In vivo fluorescence imaging of livers of a transgenic zebrafish line Tg (pck1: tagGFP-DEVD-tagRFP) at 5 dpf exposed to INH (6 mM for 24 h) with and without LPZ (10 or 50 μM for 24 h). (**C**) Hepatic apoptosis was assessed as the ratio of fluorescence signals through narrow and wide filters of the liver of zebrafish exposed to INH with or without LPZ. Black circles represent individual zebrafish, and bars represent the group medians. *n* = 10 for each group. * *p* < 0.05, ** *p* < 0.01. (**D**) The expression of *tnfa* in zebrafish exposed to INH with or without LPZ was assessed using quantitative PCR analysis. Black circles represent individual zebrafish, and bars represent the group medians. *n* = 4 for control and INH only, and *n* = 3 for INH with LPZ. * *p* < 0.05, ** *p* < 0.01.

**Table 1 pharmaceuticals-17-00082-t001:** Ontologies enriched in the GES associated with IILI.

Group	Term	Source	Associated Genes Found
Up	disulfide oxidoreductase activity	GO_MF	[PDIA3, PDIA4, PDIA6]
Up	intramolecular oxidoreductase activity	GO_MF	[PDIA3, PDIA4, PDIA6]
Up	intramolecular oxidoreductase activity, transposing S-S bonds	GO_MF	[PDIA3, PDIA4, PDIA6]
Up	oxidoreductase activity, acting on a sulfur group of donors	GO_MF	[PDIA3, PDIA4, PDIA6]
Up	protein disulfide isomerase activity	GO_MF	[PDIA3, PDIA4, PDIA6]
Up	protein disulfide reductase activity	GO_MF	[PDIA3, PDIA4, PDIA6]
Up	binding and uptake of ligands by scavenger receptors	REACTOME	[CALR, HYOU1, SAA1]
Up	IRE1alpha activates chaperones	REACTOME	[DNAJB11, HYOU1, PDIA6]
Up	Unfolded Protein Response (UPR)	REACTOME	[CALR, DNAJB11, HYOU1, PDIA6]
Up	XBP1(S) activates chaperone genes	REACTOME	[DNAJB11, HYOU1, PDIA6]
Down	bile acid biosynthetic process	GO_BP	[CES1, CYP27A1, SLC27A5]
Down	bile acid metabolic process	GO_BP	[CES1, CYP27A1, SLC27A5]
Down	cellular response to lipoprotein particle stimulus	GO_BP	[CES1, LPL, MIA3]
Down	lipid storage	GO_BP	[CES1, ENPP1, HEXB, LPL]
Down	lipopolysaccharide-mediated signaling pathway	GO_BP	[EHHADH, LBP, NFKBIL1]
Down	negative regulation of cellular response to insulin stimulus	GO_BP	[ENPP1, LPL, SOCS3]
Down	negative regulation of tumor necrosis factor production	GO_BP	[EHHADH, IGF1, LBP, NFKBIL1]
Down	negative regulation of tumor necrosis factor superfamily cytokine production	GO_BP	[EHHADH, IGF1, LBP, NFKBIL1]
Down	neuroinflammatory response	GO_BP	[CTSC, IGF1, JUN]
Down	positive regulation of chemokine production	GO_BP	[EGR1, EHHADH, LBP, LPL]
Down	positive regulation of macrophage activation	GO_BP	[CTSC, EHHADH, LBP]
Down	positive regulation of pattern recognition receptor signaling pathway	GO_BP	[EHHADH, LBP, RSAD2]
Down	positive regulation of toll-like receptor signaling pathway	GO_BP	[EHHADH, LBP, RSAD2]
Down	regulation of cellular response to insulin stimulus	GO_BP	[ENPP1, LPL, SOCS3]
Down	regulation of DNA-templated transcription in response to stress	GO_BP	[EGR1, JUN, MIA3]
Down	regulation of macrophage activation	GO_BP	[CTSC, EHHADH, LBP]
Down	regulation of release of cytochrome c from mitochondria	GO_BP	[BCL2L11, BNIP3, IGF1]
Down	regulation of toll-like receptor signaling pathway	GO_BP	[EHHADH, LBP, NFKBIL1, RSAD2]
Down	regulation of transcription from RNA polymerase II promoter in response to stress	GO_BP	[EGR1, JUN, MIA3]
Down	release of cytochrome c from mitochondria	GO_BP	[BCL2L11, BNIP3, IGF1]
Down	response to lipoprotein particle	GO_BP	[CES1, LPL, MIA3]
Down	oxidoreductase activity, acting on the CH-CH group of donors	GO_MF	[PECR, RETSAT, TM7SF2]
Down	drug metabolism	KEGG	[CES1, GSTA5, UGT2B7, UPP2]
Down	PPAR signaling pathway	KEGG	[CYP27A1, EHHADH, LPL, SCD, SLC27A5]
Down	interferon alpha/beta signaling	REACTOME	[EGR1, RSAD2, SOCS3]

UP, upregulated GES; Down, downregulated GES; GO_MF, gene ontologies in molecular function; GO_BP, gene ontologies in biological process.

**Table 2 pharmaceuticals-17-00082-t002:** Characteristics of patients treated with INH with and without LPZ.

Characteristics	Non-LPZ (*n* = 226)	LPZ (*n* = 30)	*p* Value
Female	123 (54)	13 (43)	0.258
Age (years)	66 [27–93]	71 [34–86]	0.043
Body weight (kg)	59.0 [34.5–113.9]	55.2 [35–120]	0.387
Smoking history	56 (25)	7 (23)	0.975
Drinking history	40 (18)	4 (13)	0.648
Medical history			
Heart disease	21 (9)	3 (10)	0.748
Hypertension	4 (2)	1 (3)	0.338
Type 2 diabetes disease	7 (3)	1 (3)	0.664
Biological parameters before INH treatment			
AST (U/L)	21 [7–81]	19 [6–48]	0.206
ALT (U/L)	16 [4–81]	15 [5–69]	0.550
γ-GTP	23 [8–439]	25 [8–90]	0.916
T-Bil	0.5 [0.2–3.0]	0.5 [0.2–2.1]	0.279
ALP	235 [97–1606]	221 [108–583]	0.628
LDH	189 [85–429]	180 [100–297]	0.194
Scr (mg/dL)	0.65 [0.36–1.1]	0.61 [0.38–0.92]	0.204
eGFR (mL/min)	81.1 [59.0–151]	85.7 [63.6–126.4]	0.113
Duration of INH administration (day)	186 [7–3385]	250 [21–998]	0.040
Dose of INH (mg)	250 [100–400]	250 [100–400]	0.464
Biological parameters after INH treatment			
AST (U/L)	34 [13–2424]	27.5 [15–179]	0.017
ALT (U/L)	34 [4–1550]	23.5 [8–305]	0.013
γ-GTP	30.0 [9–551]	24.0 [10–112]	0.370
T-Bil	0.7 [0.2–3.6]	0.6 [0.2–1.3]	0.059
ALP	245 [118–2214]	222 [100–400]	0.108
LDH	202 [96–837]	201.5 [115–309]	0.754
Scr (mg/dL)	0.61 [0.4–1.83]	0.58 [0.38–1.1]	0.232
eGFR (mL/min)	79.7 [28.8–223]	87.3 [46.6–155.3]	0.184

Values are presented as medians [ranges] or numbers (%). ALP, alkaline phosphatase; ALT, alanine transaminase; AST, aspartate transaminase; eGFR, estimated glomerular filtration rate; γ-GTP, γ-glut amyl trans peptidase; LDH, lactate dehydrogenase; Scr, serum creatinine; T-Bil, total bilirubin.

## Data Availability

Data is contained within the article and Appendix A.

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
