# Peer review of "Lansoprazole Ameliorates Isoniazid-Induced Liver Injury"

_pharmaceuticals, 2024, doi:10.3390/ph17010082_

Round 1

Reviewer 1 Report

Comments and Suggestions for Authors

The manuscript titled “Lansoprazole ameliorates isoniazid-induced liver injury” submitted by Wakai et al was reviewed. Herein, the authors have reported the use lanzoprazole in overcoming liver injuries induced by Isoniazid. As Isoniazid is prescribed for longer period of time, this work promises repurpose of lansoprazol. The manuscript describes scientific reasoning pertaining to gene regulation in a logical manner.

Please practice impersonation in reporting different methods and results.

Please elaborate if the same mechanism “co-treatment with LPZ significantly suppressed INH-induced activation of caspase 3 in the zebrafish liver” is admissible in human.

Sample size of groups non LPZ and LPZ described in table 2 has ~10 fold difference. How can compare the results.

Manuscript should be revised for typographic errors.

Comments on the Quality of English Language

Manuscript requires improvement in presentation in terms of Quality of English Language.

Please avoid extensive of use of terms "we".

Use of past tense in methods and results section is recommended.

Please remove typographic errors in the manuscript.

Author Response

We would like to express our gratitude to the reviewer for the constructive comments. We have revised the manuscript to address all the comments and concerns, and we believe the changes have greatly improved the manuscript. The revised text is indicated in the manuscript in red font. Below, please find our point-by-point responses to the reviewer's comments.

The manuscript titled “Lansoprazole ameliorates isoniazid-induced liver injury” submitted by Wakai et al was reviewed. Herein, the authors have reported the use lanzoprazole in overcoming liver injuries induced by Isoniazid. As Isoniazid is prescribed for longer period of time, this work promises repurpose of lansoprazol. The manuscript describes scientific reasoning pertaining to gene regulation in a logical manner.

  1. Please practice impersonation in reporting different methods and results. Please avoid extensive of use of terms "we". Use of past tense in methods and results section is recommended.

We have revised the result and method sections as suggested by the reviewer.

  1. Please elaborate if the same mechanism “co-treatment with LPZ significantly suppressed INH-induced activation of caspase 3 in the zebrafish liver” is admissible in human.

We have added the following sentences in the discussion of the revised manuscript (lines 252-258). "We showed that co-treatment of LPZ suppressed the activation of caspase 3 induced by INH in zebrafish liver. It has been demonstrated that caspase 3 was activated by INH in a human hepatocellular carcinoma line [36] and that suppression of apoptosis was involved in the hepatoprotective effects of chemicals against INH-induced injury in a human liver cell line and rat liver [37, 38]. LPZ suppressed apoptosis in an acute liver injury rat model [27]. These reports suggest that LPZ may suppress apoptosis induced by INH in the human liver."

  1. Sample size of groups non LPZ and LPZ described in table 2 has ~10 fold difference. How can compare the results.

We have added the following sentences in the revised manuscript's discussion (lines 263-267). (lines 263-267). "It should be noted, however, that the number of patients treated with INH and LPZ (n=30) is about seven times smaller than that of patients treated with INH but not LPZ (n=226). A more significant number of patients treated with INH and LPZ should be included to reduce the power imbalance and the sensitivity to outliers."

  1. Manuscript should be revised for typographic errors.
  2. Manuscript requires improvement in presentation in terms of Quality of English Language.

The revised manuscript was subjected to English proofreading.

Reviewer 2 Report

Comments and Suggestions for Authors

1.     there is no information why the zebrafish model can be used in studies of liver damage - please complete

2. Please emphasize, both in the abstract and introduction, the innovativeness of the research

Author Response

We would like to express our gratitude to the reviewer for the constructive comments. We have revised the manuscript to address all the comments and concerns, and we believe the changes have greatly improved the manuscript. The revised text is indicated in the manuscript in red font. Below, please find our point-by-point responses to the reviewer's comments.

  1. there is no information why the zebrafish model can be used in studies of liver damage - please complete

We have added the following sentences in the introduction of the revised manuscript (lines 46-50). "Zebrafish have been used for analyzing a variety of liver diseases [17-19]. Although zebrafish liver has tissue architecture different from that of humans, the cell types and the functions of zebrafish liver are similar to those of humans [17, 20]. Zebrafish have been successfully utilized to study liver injury caused by hepatotoxicants, including INH [18, 21-23]."

  1. Please emphasize, both in the abstract and introduction, the innovativeness of the research

We have revised the abstract and the introduction to emphasize the innovativeness of this study (lines 20-21 and 54-55).

Reviewer 3 Report

Comments and Suggestions for Authors

The manuscript is a good piece of work however an important fact was not discussed, what about the drug-drug interaction. It is well known that isoniazid affects the CYP system for example the 2C19 see PMID: 11158730, 11868802, etc. How this fact will be integrated in the discussion.

Comments on the Quality of English Language

non applicable

Author Response

We would like to express our gratitude to the reviewer for the constructive comments. We have revised the manuscript to address all the comments and concerns, and we believe the changes have greatly improved the manuscript. The revised text is indicated in the manuscript in red font. Below, please find our point-by-point responses to the reviewer's comments.

The manuscript is a good piece of work however an important fact was not discussed, what about the drug-drug interaction. It is well known that isoniazid affects the CYP system for example the 2C19 see PMID: 11158730, 11868802, etc. How this fact will be integrated in the discussion.

We have added the following sentences in the discussion of the revised manuscript (lines 267-271). "It should also be mentioned that INH can inhibit cytochrome P450 family 2 subfamily C member 19 (CYP2C19) [39, 40], a major phase I enzyme that metabolizes LPZ [41]. Although LPZ can be metabolized by cytochrome P450 family 3 subfamily A member 4 [41], the pharmacokinetics of LPZ co-treated with INH remain unknown."

Reviewer 4 Report

Comments and Suggestions for Authors

In the present manuscript, titled “Lansoprazole ameliorates isoniazid-induced liver injury”, Wakai and co-workers reported their investigation on the gene expression signature identification for isoniazid-induced liver injury using a public transcriptome dataset, focusing on the differences in susceptibility to isoniazid in various mouse strains. They predicted that lansoprazole is a potentially protective drug against isoniazid-induced liver injury using connectivity mapping and an adverse event reporting system. Furthermore, they confirmed the protective effects of lansoprazole against isoniazid-induced liver injury using zebrafish and electronic health records of patients. In details, their results suggest that lansoprazole can ameliorate isoniazid-induced liver injury and that the integrative approach used in this study may be applied to identify novel functions of clinical drugs.

In general, the manuscript is well organized, the methods used are appropriate, the results are clearly presented and discussed.

I think that a summary about advantages and disadvantages of your method, must be reported in the Discussion section

These minor edits will enhance impact and priority of the new method in the field.

Comments on the Quality of English Language

Minor editing of English language required

Author Response

We would like to express our gratitude to the reviewer for the constructive comments. We have revised the manuscript to address all the comments and concerns, and we believe the changes have greatly improved the manuscript. The revised text is indicated in the manuscript in red font. Below, please find our point-by-point responses to the reviewer's comments.

In the present manuscript, titled “Lansoprazole ameliorates isoniazid-induced liver injury”, Wakai and co-workers reported their investigation on the gene expression signature identification for isoniazid-induced liver injury using a public transcriptome dataset, focusing on the differences in susceptibility to isoniazid in various mouse strains. They predicted that lansoprazole is a potentially protective drug against isoniazid-induced liver injury using connectivity mapping and an adverse event reporting system. Furthermore, they confirmed the protective effects of lansoprazole against isoniazid-induced liver injury using zebrafish and electronic health records of patients. In details, their results suggest that lansoprazole can ameliorate isoniazid-induced liver injury and that the integrative approach used in this study may be applied to identify novel functions of clinical drugs. In general, the manuscript is well organized, the methods used are appropriate, the results are clearly presented and discussed.

I think that a summary about advantages and disadvantages of your method, must be reported in the Discussion section. These minor edits will enhance impact and priority of the new method in the field.

We have revised the discussion to clarify the method's advantages and disadvantages (lines 274-281). "In summary, we have demonstrated that LPZ ameliorates IILI by applying the integrative approach using in silico data, zebrafish, and real-world data, such as electronic health records and self-reporting data for adverse drug events. The integrative approach may be applicable to identify novel functions of clinical drugs leading to drug repositioning. This approach relies on the availability of in silico and real-world data and the assays using zebrafish relevant to the disease of interest. Data sharing and developing various zebrafish-based assays have grown steadily [42, 43], increasing the integrative approach's applicability."

Round 2

Reviewer 2 Report

Comments and Suggestions for Authors

Accept in present form